# S4++: Elevating Long Sequence Modeling with State Memory Reply

## Abstract

Recently, state space models (SSMs) have shown significant performance advantages in modeling long sequences. However, in spite of their promising performance, there still exist limitations. 1) Non-Stable-States (NSS): Significant state variance discrepancies arise among discrete sampling steps, occasionally resulting in divergence. 2) Dependency Bias: The unidirectional state space dependency in SSM impedes the effective modeling of intricate dependencies. In this paper, we conduct theoretical analysis of SSM from the even-triggered control (ETC) theory perspective and first propose the presence of NSS Phenomenon. Our findings indicate that NSS primarily results from the sampling steps, and the integration of multi-state inputs into the current state significantly contributes to the mitigation of NSS. Building upon these theoretical analyses and findings, we propose a simple, yet effective, theoretically grounded State Memory Reply (SMR) mechanism that leverages learnable memories to incorporate multi-state information into the current state. This enables the precise modeling of finer state dependencies within the SSM, resulting in the introduction of S4+. Furthermore, we integrate the complex dependency bias into S4+ via interactive cross attentions mechanism, resulting in the development of S4++. Our extensive experiments in autoregressive language modeling and benchmarking against the Long Range Arena demonstrate superior performance in most post-processing tasks.

## 1 Introduction

Long sequence modeling has attracted extensive interest due to its promising prospects in natural language processing, computer vision, and speech processing. The prevailing architectures for sequence modeling primarily revolve around attention-based Transformers(Vaswani et al., 2017b). However, they encountered issues like quadratic complexity (Choromanski et al., 2020; Wang et al., 2020; Beltagy et al., 2020) in the attention mechanism and computational bottlenecks related to the softmax operator(Qin et al., 2021), which constrain their effectiveness in modeling long sequences. Another perspective emerges through the framework of state space descriptions, exemplified by the State Space Model (SSM)(Gu et al., 2021a; Gupta et al., 2022; Fu et al., 2022). Among models build upon the SSM framework, S4(Fu et al., 2022) stands out in the evaluation of long sequence modeling. This is attributed to its efficient HiPPO design and matrix optimizations, which are particularly evident in the Long Range Arena (LRA) benchmark (Tay et al., 2020). The LRA benchmark focuses on comprehensive long-range modeling across diverse tasks, involving sequences ranging from 1k to 16k in various modalities.

However, S4 exhibits two inherent limitations: 1) Non Stable States (NSS). S4 simplifies the convolutions process using the Fast Fourier Transform (FFT) (Cooley & Tukey, 1965), which employs a fixed grid. However, real input information may not align perfectly with these grid points, resulting in adaptations that limit the SSM's ability to capture finer-grained states. Notably, this issue has not been recognized previously. 2) Dependency Bias. S4 conducts state transitions within one-way dependencies, which limits its effectiveness in dealing with complex dependencies, especially those commonly encountered in natural language sequences. In this paper, we provide a comprehensive response to these questions by introducing S4++.

**Q1: How does the sampling grid impact SSM?** In response to this question, we investigate SSM with various sampling grids using event-triggered control (ETC) theory (Girard, 2015). Through

this analysis, we first unveil the non stable state (NSS) issue. In a given SSM model, not all sampled sequences ensure the stability of latent states. Usually, selective sampling of input sequences is required under specific conditions for stability. Building upon this, we propose a simple yet effective the State Memory Replay (SMR) mechanism to overcome NSS. The SMR mechanism incorporates global discrete input information as a learnable memory variable, enabling grid adaptability. Using the SMR mechanism, we develop the S4+ model and confirm its ability to maintain stable latent states and can capture finer state information.

**Q2: How incorporating complexity inductive dependency bias in S4+?** To further response this question, we start by presenting an illustrative example that highlights the challenges S4 and S4+ face when capturing complex dependencies, particularly within natural language sequences. Specifically, We design a simple cloze (not unidirectional dependency) test example to assess the dependency of S4+ on the introduction of attention in capturing complex inductive biases. This motivates us to integrate the complex dependency bias into S4+ via design of cross-attention interactive, resulting in the development of S4++.

We thoroughly evaluate performance of S4++ across various sequence modeling tasks and benchmarks, including autoregressive language modeling, text classification and the Long Range Arena benchmark. S4++ demonstrates superior performance compared to a variety of state of the art baseline models.

## 2 THE NON-STABLE PROBLEM IN STATE SPACE MODEL

In this section, we primarily introduce the State Space Model (SSM) and then provide a detailed analysis of Non-Stable-States (NSS) within ETC Theory. Specifically, we discuss SSM in Section 2.1 and explain the causes and phenomenon of NSS in Sections 2.2 and 2.3.

### 2.1 STATE SPACE MODEL

The state space model is formally defined by equations 1 and 2:

$$\dot{\boldsymbol{x}}(t) = \boldsymbol{A}\boldsymbol{x}(t) + \boldsymbol{B}u(t) \tag{1}$$

$$y(t) = \boldsymbol{C}\boldsymbol{x}(t) + \boldsymbol{D}u(t) \tag{2}$$

where $\boldsymbol{A} \in \mathbb{R}^{n \times n}$, $\boldsymbol{B} \in \mathbb{R}^{n \times m}$, $\boldsymbol{C} \in \mathbb{R}^{m \times n}$, $\boldsymbol{D} \in \mathbb{R}^{m \times m}$, $u(\cdot) : \mathbb{R} \mapsto \mathbb{R}^m$ denotes the input sequence with dimension $m$, and $\boldsymbol{x}(\cdot) : \mathbb{R} \mapsto \mathbb{R}^n$ is the latent state. Previous work (Gu et al., 2021b; 2022) formed the so-called S4 model and undergoes discretization using the bilinear method to adapt it to discrete inputs:

$$\boldsymbol{x}_k = \overline{\boldsymbol{A}}\boldsymbol{x}_{k-1} + \overline{\boldsymbol{B}}u_k \tag{3}$$

$$y_k = \overline{\boldsymbol{C}}\boldsymbol{x}_k, \tag{4}$$

where $\overline{\boldsymbol{A}} = (\boldsymbol{I} - \Delta t/2 \cdot \boldsymbol{A})^{-1}(\boldsymbol{I} + \Delta t/2 \cdot \boldsymbol{A})$, $\overline{\boldsymbol{B}} = (\boldsymbol{I} - \Delta t/2 \cdot \boldsymbol{A})^{-1}\Delta \boldsymbol{B}$, $\overline{\boldsymbol{C}} = \boldsymbol{C}$, $\Delta t$ is the discrete step size the same for all steps. The matrix $D$ is omitted here because it can be viewed as a residual connection. Then, the S4 becomes a parameterized model with trainable parameters $\overline{\boldsymbol{A}}$, $\overline{\boldsymbol{B}}$, $\overline{\boldsymbol{C}}$, and $\Delta t$. By assuming $x_0 = \boldsymbol{0}$, we can obtain:

$$y_k = \overline{\boldsymbol{C}\boldsymbol{A}}^{k-1}\overline{\boldsymbol{B}}u_1 + \overline{\boldsymbol{C}\boldsymbol{A}}^{k-2}\overline{\boldsymbol{B}}u_2 + \cdots + \overline{\boldsymbol{C}\boldsymbol{A}\boldsymbol{B}}u_{k-1} + \overline{\boldsymbol{C}\boldsymbol{B}}u_k, \tag{5}$$

thus the output could be calculated efficiently by FFT $y = \overline{\boldsymbol{K}} * u$, where

$$\overline{\boldsymbol{K}} \in \mathbb{R}^L := \mathcal{K}_L(\overline{\boldsymbol{A}}, \overline{\boldsymbol{B}}, \overline{\boldsymbol{C}}) := \left(\overline{\boldsymbol{C}\boldsymbol{A}^i\boldsymbol{B}}\right)_{i \in [L-1]} = \left(\overline{\boldsymbol{C}\boldsymbol{B}}, \overline{\boldsymbol{C}\boldsymbol{A}\boldsymbol{B}}, \ldots, \overline{\boldsymbol{C}\boldsymbol{A}^{L-1}\boldsymbol{B}}\right) \tag{6}$$

is the convolution kernel and $L$ is the sequence length.

### 2.2 NON-STABLE-STATES PHENOMENON

Through the ETC theory, we provide a simple example to elucidate the phenomenon of non-stable states (NSS). In this context, ETC ensures the system's states remain stable by sampling the input

control signal using triggered events. To maintain stability, the selection of sampling points, such as $t_1, t_2, \ldots$, must meet specific criteria. Typically, a Lyapunov function $\mathcal{L}_V$ is employed to assess stability (Heemels et al., 2012), outside the stable point, it is monotonically decreasing, and the minimum value of $0$ is achieved at the stable point. Sampling points that result in a decreasing trend of $\mathcal{L}_V$ are selected to ensure system stability. Specifically, consider the linear system described in eq. 1. Assuming the input control signal satisfies the linearity $u(t) = \boldsymbol{T}\boldsymbol{x}(t)$, where $\boldsymbol{T} \in \mathbb{R}^{m \times n}$, then eq. 1 becomes:

$$\dot{\boldsymbol{x}}(t) = \boldsymbol{A}\boldsymbol{x}(t) + \boldsymbol{B}\boldsymbol{T}\boldsymbol{x}(t). \tag{7}$$

It can be easily verified that $\mathcal{L}_V(t) = \boldsymbol{x}^T \boldsymbol{P} \boldsymbol{x}$ is a Lyapunov function, where symmetric positive definite matrix $\boldsymbol{P} \in \mathbb{R}^{n \times n}$ satisfies:

$$(\boldsymbol{A} + \boldsymbol{B}\boldsymbol{T})^\top \boldsymbol{P} + \boldsymbol{P}(\boldsymbol{A} + \boldsymbol{B}\boldsymbol{T}) = -\boldsymbol{M}, \tag{8}$$

where $\boldsymbol{M} \in \mathbf{R}^{n \times n}$ is also a symmetric positive definite matrix. Note that the actual sampled input $u(t_i)$ is sampled at the sampling points $\{t_i\}_{i \in \mathbb{N}}$, we denote the sampling error:

$$\boldsymbol{e}(t) = \boldsymbol{x}(t_i) - \boldsymbol{x}(t), \quad \forall t \in [t_i, t_{i+1}), i \in \mathbb{N}, \tag{9}$$

then eq 7 could be reformulated as:

$$\dot{\boldsymbol{x}}(t) = \boldsymbol{A}\boldsymbol{x}(t) + \boldsymbol{B}\boldsymbol{T}(\boldsymbol{x}(t) + \boldsymbol{e}(t)). \tag{10}$$

Taking the derivative of $\mathcal{L}_V$, we have:

$$\frac{d}{dt}\mathcal{L}_V(t) = -\boldsymbol{x}(t)^T \boldsymbol{M}\boldsymbol{x}(t) + 2\boldsymbol{x}(t)^\top \boldsymbol{P}\boldsymbol{B}\boldsymbol{T}\boldsymbol{e}(t). \tag{11}$$

Therefore, set $t_0 = 0$, we have the following triggering condition to ensure system stability:

$$t_{i+1} = \inf \left\{ t \in \mathbb{R} \mid t > t_i \wedge \kappa \boldsymbol{x}(t)^\top \boldsymbol{M}\boldsymbol{x}(t) - 2\boldsymbol{x}(t)^\top \boldsymbol{P}\boldsymbol{B}\boldsymbol{T}\boldsymbol{e}(t^-) \leq 0 \right\} \tag{12}$$

where $\kappa \in (0, 1)$ is a optional constant, $e(t^-)$ represents the left-hand limit of error $e$ at point $t$. In this way, the sampled input control sequence obtained can ensure exponential stability of the system:

$$\mathcal{L}_V(t) \leq \mathcal{L}_V(0)e^{(\kappa-1)\iota t}, \tag{13}$$

where $\iota$ is an positive constant. More specifically, we provide an example of a 1-D input where the selected parameters are as follows:

$$\boldsymbol{A} = \begin{bmatrix} 0 & 1 \\ 2 & -3 \end{bmatrix}, \boldsymbol{M} = \begin{bmatrix} 0.5 & 0.25 \\ 0.25 & 1.5 \end{bmatrix}, \boldsymbol{P} = \begin{bmatrix} 1 & 0.25 \\ 0.25 & 1 \end{bmatrix}, \boldsymbol{B} = \begin{bmatrix} 0 \\ 1 \end{bmatrix}, \boldsymbol{T} = \begin{bmatrix} 1 \\ -4 \end{bmatrix},$$

The selected time window is $[0, 10]$, with a time grid width of $0.01$. Subsequently, we conduct simulation experiments on the system, and the results is shown in the leftmost of Fig.1, the triggering moment is marked with a gray dashed line. Under the sampled input obtained from ETC, the system's state eventually reaches the stabilization.

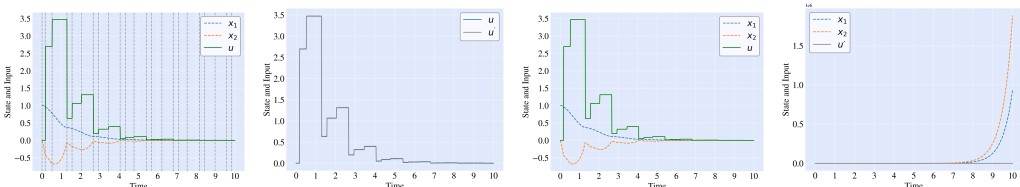

Figure 1: An example of the issue of NSS in SSM.

**NSS: Instability Arising from sampling Grid Mismatch.** To further substantiate this conclusion, we present an illustrative example. Specifically, we introduce minor perturbations to the sampled data points, strictly constrained within the temporal grid width. The comparison between the perturbed input and the original input is illustrated in the second figure in Fig.1, where the perturbations exhibit almost imperceptible variances. When utilizing the unaltered sampled data points obtained prior to perturbation as input, the third figure visually represents the system's sustained stability. Nevertheless, upon the introduction of perturbated sampled data points into the system, as depicted in the rightmost in Fig.1, it becomes apparent that the system's stability cannot be guaranteed, leading to an exponential growth in magnitude reaching $10^6$. This means that when the actual sampling points do not align with the desired sampling grid, it will result in highly unstable states. For S4, if it also encounters such an issue, it will inevitably lead to unavoidable numerical errors ( Proposition 1).

## 2.3 NSS Effect On S4

Based on the aforementioned considerations and insights, our understandings of S4 are enlightened as follows: (1) S4 maintains a uniform $\Delta t$ value across different positions, but it doesn't guarantee consistent system stability when using the same $\Delta t$. (2) S4 does not implement specific adjustments for state stability, which can result in unstable states even during training data processing. (3) During the inference stage, the S4 model ensures consistent parameter settings, especially concerning the sampling grid, which may introduce a certain degree of inference error. Once the latent state experiences exponential growth, it inevitably leads to numerical errors, as affirmed by Proposition 1.

**Proposition 1** *Given bounded inputs satisfying $\|u\| \leq \zeta$ and $\|\boldsymbol{B}\| < b$, and defining the observation error caused by sampling points as $\boldsymbol{\varepsilon}_i = u_i' - u_i$, it can be concluded that when $\lim_{t \to \infty} \|\boldsymbol{x}_t\| > \frac{b\zeta}{1 - |\lambda_{\max}|}$, where $\lambda_{\max}$ represents the largest eigenvalue of matrix $\overline{\boldsymbol{A}}$, the prediction error $\|y_t' - y_t\|$ will over time steps.*

To ascertain the presence of an NSS issue within the S4 model, we devise a simple sequence modeling task. We sampled 100 equidistant points from the function $\sin(5\pi t)$ to serve as input $u$. Then, we employ a single-layer S4 model for fitting, which underwent training for 2000 epochs, yielding the results displayed in the leftmost in Fig.2. Following a methodology akin to the previous example, we apply perturbations smaller than the sampling window width to the sampled points $\{t_i\}_{i \in [99]}$. Subsequently, we conduct sampling on the perturbed points $\{t_i'\}_{i \in [99]}$. This process generated a set of perturbed inputs, denoted as $u'$, as illustrated in the second figure in Fig.2, where the sampling points underwent slight alterations. Subsequently, we employ the trained S4 model to predict $u'$, resulting in a numerical instability, as evident in the third figure in Fig.2. We graphically represent the latent states before and after perturbation in the rightmost figure in Fig.2. In both instances, unstable states were observed, and notably, the total magnitude of the state increased following the perturbation. We extend this verification to a 5-layer S4 model and observe analogous findings. The outcomes are detailed in Appendix A.2. The non-uniform sampling point variations give rise to the NSS problem, which not only induces numerical instability and constrains the model's generalization capability but also leads to unstable gradients, thus impeding the model's fitting capacity, as demonstrted in the third and last figure in Fig. 4. Therefore, improving the NSS problem holds the potential to enhance the fitting and generalization capabilities of SSM-based models.

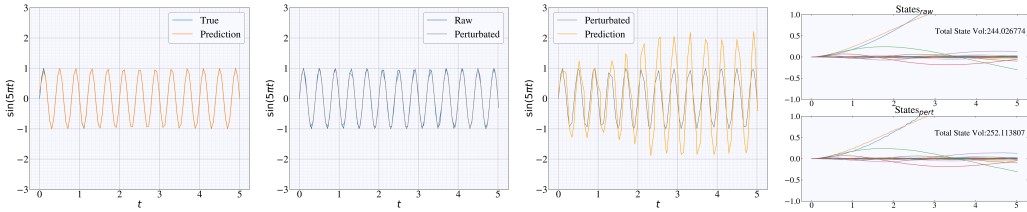

Figure 2: An illustrative instance of the NSS issue in S4 is presented here. States$_{raw}$ and States$_{Pert}$ denote the latent states of the model when applied to the original fitting data and the data subject to sampled perturbations, respectively. The Total State Vol signifies the summation of absolute values across all states at each time point.

## 3 S4++

In this section, we commence with a theoretical analysis from the perspective of ETC theory. We discover that the inclusion of multiple states in the current state has a positive impact on divergence mitigation. Expanding upon these comprehensive analyses and profound insights, we propose a State Memory Replay (SMR) mechanism aimed at addressing the NSS problem caused by variations in sampling points. The SMR mechanism incorporates learnable memories to enhance the S4 framework with multiple states, as depicted in Fig.3, resulting in the introduction of S4+. Subsequently, we devise a series of experiments to validate the effectiveness of the proposed SMR mechanism. Subsequently, we incorporate an intriguing example to offer insights into why S4+ may

not excel in language modeling tasks, focusing on its language modeling capabilities (3.2). Furthermore, we explore the idea of improving S4+ by adding cross attention mechanisms, leading to the development of S4++ (3.3).

### 3.1 S4+: S4 WITHIN SMR MECHANISM

We start by offering a simplified theoretical explanation for the necessity of incorporating the SMR mechanism into the S4 model. We examine an input perturbation denoted as $\varepsilon$ at the sampling point, where $u(t + t_\varepsilon) = u(t) + \dot{u}(t)t_\varepsilon + o(t_\varepsilon)$. Assuming a tiny perturbation $\varepsilon(t)$, we have $u'(t) = u(t) + \varepsilon(t)$. Hence, the observed state $z(t)$ can be expressed as $u'(t) = u(t) + \varepsilon(t)$, and we also define the discrepancy between the observed state and the actual state as the error $e(t) = x(t) - z(t)$. Drawing inspiration from EMC theory, the Lyapunov function L is utilized as an indicator of observation error stability in the system. A smaller absolute value of $e(t)$ indicates a reduced impact of noise and uncertainty on system performance, as demonstrated in (Vallarella & Haimovich, 2019). Then, we have Theorem 1.

**Theorem 1** *For the input reply factor $h_\tau(t) = h([t - \tau, t]) : [t - \tau, t] \to \mathbb{R}$, the adjusted input $u_{adj}(t) = h_\tau(t)u(t)$, where $z(t)$ is the state value of observer, considering the Lyapunov function $\mathcal{L}_e(t) = e^\top(t)Pe(t)$, we have:*

$$\frac{d\mathcal{L}_e(t)}{dt} \leq e^\top(t)\left(PA + A^\top P\right)e(t) + \|h_\tau\|_\infty \|e(t)\| \left(\int_0^t \|k(t - l)\| |\varepsilon(l)| \, dl + \|B\| |\varepsilon(t)|\right),$$
(14)

*where $P$ is a positive definite symmetric matrix and $k(\cdot) : \mathbb{R} \to \mathbb{R}^n$ is a coefficient function.*

**Remark 1.** Theorem 1 suggests that imposing additional constraints on the input controller $h_\tau$ can improve the convergence of the system. In particular, when $h_\tau(\cdot) \equiv 1$ (denoted as S4), we have $\|h_\tau\|_\infty = 1$. The control factor $h_\tau$ is required to incorporate information from the time interval $[t - \tau, t]$. To accomplish this, a convolution $\text{Conv}_\tau$ with a kernel of length $\tau$, denoted as $\mathcal{K}_\tau$, can be utilized. Moreover, an activation function, denoted as $\sigma$, can be employed to ensure that the condition $\|h_\tau\|_\infty = \|\sigma \circ \text{Conv}_\tau\|_\infty < 1$ is satisfied. This condition contributes to the enhancement of system stability.

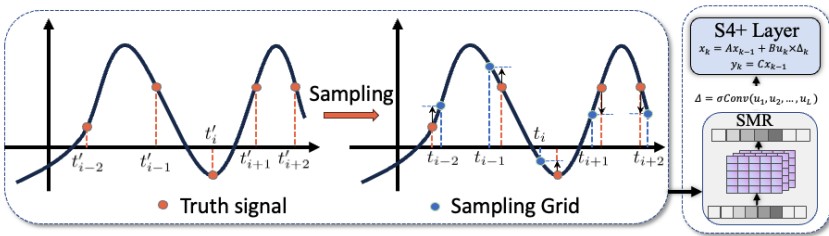

Figure 3: Illustration of the proposed SMR Mechanism.

Building upon Theorem 1, we understand the importance of having learnable variables that can incorporate multi input states to control how sampling information behaves, allowing for automatic adjustments. To meet this need, considering the analysis in **Remark 1**, we propose the design of a convolutional learnable variables that incorporates multi input states, enabling adaptive learning and refinement. Formally, our proposed SMR mechanism can be formulated as:

$$x_k = \overline{A}x_{k-1} + \overline{B}u_k\sigma_{\text{Sig}}(\mathcal{K}_\tau * (\underbrace{u_1, \ldots, u_1}_{\tau-1}, u_1, u_2 \ldots, u_T))_k,$$
(15)

where $\tau$ represents the convolutional kernel size, and $\sigma_{\text{Sig}}(\cdot)$ refers to the Sigmoid function. This mechanism facilitates the model in making compensatory adjustments to the step size, guided by local input information, thus accomplishing the objective of alleviating the NSS problem. Naturally, with the help of this mechanism, we introduce a new model named S4+, illustrated in Fig.3. This model smoothly addresses the NSS problem by being guided by the SMR. To further assess the efficacy of the SMR mechanism in S4+, we conduct training and testing on a 1-layer S4+ model using

the same configurations as mentioned previously. The model's fitting results on $u'$ are displayed in the leftmost of Fig.4, demonstrating the successful mitigation of unstable numerical outputs and a substantial reduction in prediction errors. The results illustrated in the second figure of Fig.4 clearly indicate that the latent states of S4+ have achieved stability, characterized by a significantly reduced total volume of the absolute state values, shrinking from $2 \times 10^2$ to $3 \times 10^1$. This implies a notable improvement in addressing the NSS problem. Furthermore, we graph the sum of loss and loss gradients throughout the training process for both scenarios, with and without the SMR mechanism, as depicted in the last two figures of Fig.4. Obviously, the SMR mechanism is contributed to stabilizing the training gradients, accelerating convergence, further diminishing fitting errors, and enhancing the S4+ overall fitting capacity.

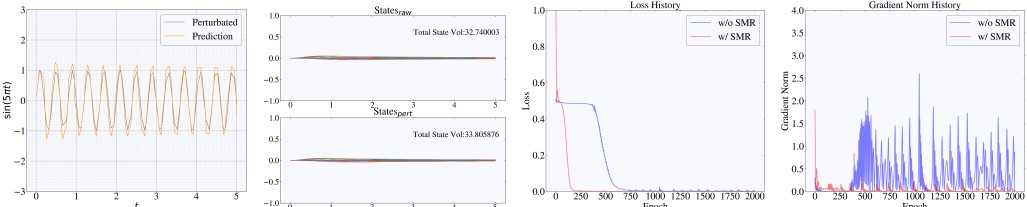

Figure 4: We conduct a comparative validation of S4+ on the aforementioned examples. The left pair of figures displays the prediction outcomes of S4+ for the perturbed input $u'$ and the latent states of the model when provided with inputs $u$ and $u'$. The right pair of figures display the training loss of the model both with and without SMR mechanism, in addition to tracking the changes in the sum of parameter gradients during the training phase.

## 3.2 WHY S4+ NEEDS ATTENTION

While SSM excels in modeling long-term dependencies, it falls short in language modeling tasks (Wang et al., 2022; Fu et al., 2023). This similarly raise concerns regarding S4+. Drawing inspiration from prior research, incorporating attention mechanisms into S4+ might offer a promising avenue. With this motivation, we began by designing a toy example to understand why SSM-based models require attention. When considering the modeling of a specific relationship: $f(y_i|u_{i-l_1}, u_{i+l_2})$, the SSM model, parameterized by $\theta$, is capable of modeling $f_\theta(y_i|u_1, \ldots, u_{i-1})$, thereby enabling the modeling of $f_\theta(y_i|u_{i-l_1})$. However, these one-way dependencies limit its ability to model $f_\theta(y_i|u_{i-l_1}, u_{i+l_2})$, which in turn restricts its capacity to capture complex semantic relationships. For instance, take the following cloze question as an example: "I got a score of A, I am the ___ student in my class because Mike got a B and Katty got a C." The correct

| | Test Error (MSE) |
|---|---|
| $f_\varphi$ | 1.19 |
| $f_\theta$ | 0.67 |
| $f_{\varphi,\theta}$ | 0.02 |

Table 1: Results of the designed experiment.

answer is 'top.' However, SSM falls short in performing such modeling tasks. Consider a Cross-Attention structure defined by the parameters $\varphi = \boldsymbol{W}_q, \boldsymbol{W}_k, \boldsymbol{W}_v$, as shown below:

$$f_\varphi(u, y) = \sigma_{\text{Soft}}((\boldsymbol{W}_q u)(\boldsymbol{W}_k y)^\top)\boldsymbol{W}_v u, \tag{16}$$

where $y = (y_1, y_2, \ldots, y_L)$ represents the output of SSM and $u = (u_1, u_2, \ldots, u_L)$ represents the input sequence. Because $f_\varphi$ captures global information by globally comparing $y$ and $u$, even a simple linear combination, denoted as $f_{\varphi,\theta} = c_1 f_\theta + c_2 f_\varphi$, can effectively model $f_{\varphi,\theta}(y|u_{i-l_1}, u_{i+l_2})$. To substantiate this viewpoint, we devise an illustrative toy dataset. Each input comprises a sequence of length 100, with values at positions 12 and 94 sampled from a uniform distribution $U[1, 2]$. Our objective is for the model to acquire knowledge of the 'Pythagorean theorem' at position 63: This example strongly supports our viewpoint. To ensure the model can capture long-term dependencies and complex semantic relationships, a combination of attention-based structures and SSM structures is essential.

$$f(y_{63}|u_{12}, u_{94}) = u_{12}^2 + u_{94}^2. \tag{17}$$

We randomly generate 200 samples and divide them into training and testing sets with an $8 : 2$ ratio. Three one-layer models were trained for 100 epochs: one with attention only ($f_\varphi$), one with S4 only

($f_\theta$), and their combination ($f_{\varphi,\theta} = f_\theta + f_\varphi$). The testing set errors (MSE) are presented in Table 1. It is observed that $f_\varphi$ struggles to adequately capture long-term dependencies, whereas $f_\theta$ faces limitations in modeling intricate dependencies. Among the models considered, only $f_{\varphi,\theta}$ demonstrates the capacity to effectively capture long-term dependencies and model complex dependencies, exhibiting accurate learning of the designed relationships.

### 3.3 S4++: Integrating the complexity dependency bias into S4+ via attention mechanisms

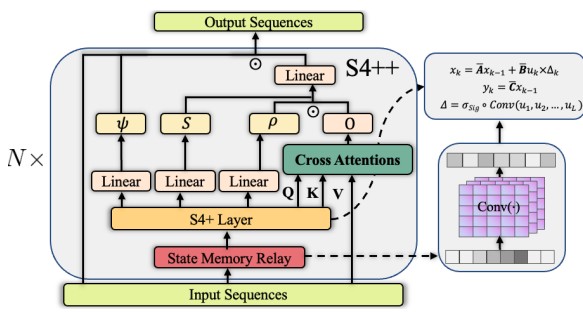

Figure 5: Illustration of the S4++ layer.

Based on the preceding analysis, our aim is to incorporate complex dependency biases into S4+. As a result, we introduce S4++ as depicted in Figure 5. Specifically, for an input $\boldsymbol{U} \in \mathbb{R}^{L \times d}$, we initially modify the input by applying Equation 15, resulting in

$$\boldsymbol{U}_{\mathrm{a}} = U \odot \sigma_{\mathrm{Sig}} \left( \mathcal{K}_\tau * \boldsymbol{U}_{\mathrm{l}_{\tau-1}} \right), \quad (18)$$

where $\boldsymbol{U}_{\mathrm{lpad}_{\tau-1}} \in \mathbb{R}^{(L+\tau-1) \times d}$ denotes the leftpadding operation with length $\tau - 1$. Then, utilizing the $\overline{\boldsymbol{K}}$ from Eq. 5, we calculate the output of S4+ as

$$\boldsymbol{U}_{\mathrm{S4+}} = \overline{\boldsymbol{K}} * \boldsymbol{U}_{adj} \quad \in \mathbb{R}^{L \times d}, \quad (19)$$

According to Equation 16, we proceed with a cross-attention operation between the output of S4+ and the original input as

$$\boldsymbol{Q} = \sigma_{\mathrm{Silu}} \left( \boldsymbol{U}\boldsymbol{W}_q + b_q \right) \qquad \in \mathbb{R}^{L \times q} \qquad (20)$$

$$\boldsymbol{K} = \sigma_{\mathrm{Silu}} \left( \boldsymbol{U}_{\mathrm{S4+}}\boldsymbol{W}_k + b_k \right) \qquad \in \mathbb{R}^{L \times k} \qquad (21)$$

$$\boldsymbol{V} = \sigma_{\mathrm{Silu}} \left( \boldsymbol{U}\boldsymbol{W}_v + b_k \right) \qquad \in \mathbb{R}^{L \times v} \qquad (22)$$

$$\boldsymbol{O} = \sigma_{\mathrm{Soft}} \left( \boldsymbol{Q}\boldsymbol{K}^T / \sqrt{d} + b_{\mathrm{pos}} \right) \boldsymbol{V} \qquad \in \mathbb{R}^{L \times v} \qquad (23)$$

Here, $\sigma_{\mathrm{silu}}$ and $\sigma_{\mathrm{Soft}}$ represent the Silu and Softmax activation functions, respectively. $\boldsymbol{W}_q, \boldsymbol{W}_k \in \mathbb{R}^{d \times q}$, $\boldsymbol{W}_v \in \mathbb{R}^{d \times v}$, $b_q, b_k \in \mathbb{R}^q$, $b_v \in \mathbb{R}^v$ are trainable parameters. $b_{\mathrm{pos}}$ denotes the positional bias. Then combine the sequences after cross attention $\boldsymbol{O}$ and after applying SSM $\boldsymbol{U}_{\mathrm{S4+}}$ in the following manner:

$$\rho = \sigma_{\mathrm{Silu}}(\boldsymbol{U}_{\mathrm{S4+}}\boldsymbol{W}_\rho + b_\rho) \qquad \in \mathbb{R}^{L \times v} \qquad (24)$$

$$\boldsymbol{S} = \boldsymbol{U}_{\mathrm{S4+}}\boldsymbol{W}_s + b_s \qquad \in \mathbb{R}^{L \times d} \qquad (25)$$

$$\boldsymbol{H} = \sigma_{\mathrm{Silu}} \left( (\rho \odot \boldsymbol{O}) \boldsymbol{W}_h + \boldsymbol{S} \right) \qquad \in \mathbb{R}^{L \times d} \qquad (26)$$

where parameters $\boldsymbol{W}_\rho \in \mathbb{R}^{d \times v}$, $\boldsymbol{W}_h \in \mathbb{R}^{v \times d}$, $b_\rho \in \mathbb{R}^v$, $b_s \in \mathbb{R}^d$. Note that $(\rho \odot \boldsymbol{O}) \boldsymbol{W}_h + S$ corresponds to the sequence after S4+ Layer and the linear combination of cross attention. Next, we utilize $\boldsymbol{U}_{\mathrm{S4+}}$ to construct a gating mechanism for the final gated residual connection:

$$\psi = \sigma_{\mathrm{Sigmoid}} \left( \boldsymbol{U}_{\mathrm{S4+}}\boldsymbol{W}_\psi + b_\psi \right) \qquad \in \mathbb{R}^{n \times d} \qquad (27)$$

$$\boldsymbol{Y} = \psi \odot H + (1 - \psi) \odot \boldsymbol{U} \qquad \in \mathbb{R}^{n \times d} \qquad (28)$$

where parameters $\boldsymbol{W}_\psi \in \mathbb{R}^{d \times d}$, $b_\psi \in \mathbb{R}^d$. The experiments in Section 4 confirm the effectiveness of our strategy in improving both language modeling (Section 4.1, 4.2) and long sequence modeling (Section 4.3) in S4++. This supports our claim and validates the enhancements in the structure of S4++.

## 4 Experiments

We conduct comparative experiments on multiple benchmark datasets to demonstrate the effectiveness of our proposed S4+ and S4++. In our experiments, we evaluate the autoregressive language

modeling capability (Section 4.1) using the WikiText-103 dataset (Merity et al., 2017). We also assess the bidirectional language modeling ability (Section 4.2) on the GLUE benchmark dataset (Wang et al., 2019). Furthermore, we validate the model's capability to learn long-range dependencies (Section 4.3) using the Long Range Arena (LRA) dataset (Tay et al., 2021).

Regarding the models under study, we compare S4+, S4++ with various types of models, including MLP-based model gMLP (Liu et al., 2021), FFT-based model GFNet (Rao et al., 2021), SSM-based model S4 (Gu et al., 2022), MEGA (Ma et al., 2023), H3 (Fu et al., 2023), TNN (Qin et al., 2023), as well as Attention-Based models Vanilla transformer (Vaswani et al., 2017a), Transformer LS (Zhu et al., 2021), FLASH (Hua et al., 2022), and cosformer (Qin et al., 2022). All experiments were conducted on four Tesla A800 GPUs.

## 4.1 AUTOREGRESSIVE LANGUAGE MODELING

To evaluate the ability of autoregressive language modeling, we conducted experiments on the WikiText-103 dataset. This dataset comprises 103 million word-level tokens extracted from Wikipedia articles. In accordance with (Qin et al., 2023), all models were trained on the WikiText-103 dataset for 50,000 steps, using a learning rate of $5e - 4$. The sequence length was set to $512$, and weight decay was set at $0.2$ for TNN and $0.1$ for other models. The performance of autoregressive language modeling is assessed by reporting perplexity (PPL) scores on both the validation and test sets. In order to ensure fairness, the parameter sizes of all models were adjusted to a comparable scale. For more detailed information regarding the experiments, please refer to Appendix A.3.

As shown in Table 2, the experimental results indicate that both S4+ and S4++ outperform other models in terms of their autoregressive language modeling abilities, including the dominant Transformer model for sequence modeling and other Attention-based SSMs such as H3 (Fu et al., 2023), MEGA (Ma et al., 2023) and TNN (Qin et al., 2023) Compared to S4, both S4+ and S4++ demonstrate substantial improvements in autoregressive language modeling. This validates the effectiveness of our proposed SMR mechanism, which enhances the model's fitting ability.

Table 2: Result on Wikitext-103, the best results are represented in bold, and the second best results are marked with an underline.

| Model | PPL(val) | PPL(test) | Params(M) |
|---|---|---|---|
| Trans | 25.51 | 25.76 | 48.12 |
| LS | **24.21** | **24.45** | 47.89 |
| Cosformer | 26.91 | 27.53 | 44.65 |
| FLASH | 25.92 | 26.70 | 42.17 |
| GMLP | 26.55 | 27.19 | 47.83 |
| H3 | 28.23 | 29.13 | 48.57 |
| TNN | 25.49 | 25.35 | 48.36 |
| MEGA | 26.30 | 26.75 | 48.68 |
| S4 | 39.86 | 40.02 | 45.69 |
| S4+(ours) | 38.14 | 38.62 | 48.57 |
| S4++(ours) | 25.24 | 25.31 | 49.85 |

## 4.2 BIDIRECTIONAL LANGUAGE MODELING

We train a bidirectional language model on the GLUE dataset to evaluate its performance across various downstream natural language understanding tasks. As described in the literature (Qin et al., 2023), we chose Roberta as the underlying model architecture for all of our methods. Initially, all models underwent pretraining on the WikiText-103 dataset, with a learning rate of $5e - 4$ applied over a span of 50,000 training steps. After pretraining, we choose the best checkpoint for fine-tuning and evaluating the model on the GLUE dataset. For each task within GLUE, we conduct training for 10 epochs, with learning rates determined in accordance with the guidelines provided in [1]. We employ a learning rate of $2e - 5$ for most datasets, while exception were made for RTE, where the learning rate was set to $1e - 5$. To maintain consistency, weight decay was set at $0.2$ for TNN and $0.1$ for other models. Additionally, we ensure that all model parameters were adjusted to similar magnitudes for fair evaluation.

As the results shown in Table 3, the language model relies more on complex interactions at a global granularity, which is exactly attention excel at, while SMR helps the SSM model capture finer-grained interactions, particularly enhancing long-term memory (4.3). Compared to S4+, the S4++ model, with the inclusion of cross-attention, has the ability to capture both global and fine-grained interactions, thereby exhibiting excellent performance across different tasks (Table 3, 4).

---

[1] https://github.com/facebookresearch/fairseq/blob/main/examples/roberta/README.glue.md

Table 3: Performances comparison of bidirectional sequence modeling on the GLUE benchmark. MNLI is reported by the match/mismatch splits, MRPC is reported by F1 score and CoLA is reported by Matthews correlation coefficient. The best results are represented in bold, and the second best results are marked with an underline.

| Model | MNLI | QNLI | QQP | SST-2 | MRPC | CoLA | RTE | AVG | Params(m) |
|---|---|---|---|---|---|---|---|---|---|
| Transformer | 78.82/**78.50** | 86.71 | 88.50 | 90.17 | **86.68** | 37.07 | 58.89 | 75.67 | 124.30 |
| LS | 73.75/73.82 | 84.63 | 85.43 | 87.07 | 82.25 | 40.67 | **62.16** | 73.72 | 128.28 |
| FLASH | 77.49/77.90 | 83.27 | 85.23 | 86.04 | 82.24 | 28.63 | 57.09 | 72.23 | 127.12 |
| GMLP | 71.54/71.85 | 77.36 | 85.69 | 81.71 | 81.91 | 35.93 | 55.89 | 70.24 | 131.08 |
| GFNet | 65.75/66.80 | 65.50 | 79.53 | 83.45 | 81.23 | 9.29 | 55.81 | 63.42 | 121.57 |
| S4 | 67.46/68.77 | 69.94 | 82.92 | 84.33 | 82.59 | 21.17 | 54.62 | 66.48 | 126.57 |
| MEGA | 76.47/76.78 | 85.73 | 88.35 | 86.93 | 82.63 | 38.36 | 57.81 | 74.13 | 130.90 |
| H3 | 75.45/75.22 | 82.55 | 85.18 | 84.86 | 82.30 | 32.10 | 57.10 | 71.85 | 136.91 |
| TNN | 75.26/75.62 | 84.48 | 86.81 | 88.01 | 82.29 | **47.21** | 57.62 | 74.66 | 126.40 |
| S4+ (ours) | 65.92/66.24 | 72.21 | 83.43 | 82.47 | 81.91 | 21.17 | 55.81 | 66.15 | 128.81 |
| S4++ (ours) | **78.94**/78.06 | **87.01** | **89.73** | **90.99** | 85.22 | 42.21 | 60.71 | **76.61** | 134.11 |

### 4.3 LONG-RANGE DEPENDENCY MODELING

To evaluate the ability of long sequence modeling, we conduct additional experiments on five LRA tasks. These tasks include ListOps (Nangia & Bowman, 2018), byte-level text classification (Maas et al., 2011), byte-level document retrieval (Radev et al., 2013), sequence CIFAR-10 (Krizhevsky & Hinton, 2009), and Pathfinder (Linsley et al., 2018). To ensure a fair comparison, we train each model with similar parameters and settings. Taking into account that RetNet (Sun et al., 2023) is a special SSM-based model, we also tested its performance on LRA. In the results presented in Table 4, S4+ exhibits a significant performance improvement compared to S4, which matched with our analysis, validating the significant assistance provided by our proposed SMR in modeling long sequences. S4++ also performing well, even compared to the current state-of-the-art baselines S4 (Fu et al., 2022), MEGA (Ma et al., 2023), and TNN (Qin et al., 2023). The average performance of S4+ and S4++ is almost identical. This is because SMR helps the SSM model capture fine-grained information more accurately, while incorporating attention is aimed at further adapting S4+ to a wider range of data patterns. Taking into account the experimental results mentioned above, it can be concluded that the two mechanisms in S4++ serve different purposes and do not restrict each other. This allows S4++ to adaptively handle different data patterns, leading to good performance across various tasks.

Table 4: Performances Comparison on the Long Range Arena benchmark. We use bold and underline to highlight the best and the second result of each task respectively. The best results are represented in bold, and the second best results are marked with an underline.

| Model | Text | ListOps | Retrieval | Image | Pathfinder | AVG |
|---|---|---|---|---|---|---|
| Transformer | 61.95 | 38.37 | 80.69 | 65.26 | 40.57 | 57.37 |
| cosFormer | 67.70 | 36.50 | 83.15 | 71.96 | 62.11 | 51.23 |
| Reformer | 62.93 | 37.68 | 78.99 | 48.87 | 66.49 | 58.99 |
| TNN | 82.74 | 42.70 | 82.97 | 70.48 | 76.54 | 71.09 |
| MEGA | 85.86 | 53.30 | 76.43 | 75.47 | **80.97** | 74.41 |
| H3 | 78.61 | 39.20 | 75.67 | 80.27 | 75.31 | 69.81 |
| RetNet | 75.58 | 42.30 | 75.08 | 77.70 | 75.23 | 69.18 |
| S4 | 86.46 | 49.75 | 84.64 | 81.70 | 72.07 | 75.32 |
| S4+(ours) | **89.23** | 51.95 | **88.37** | **84.45** | 77.86 | **78.37** |
| S4++(ours) | 86.28 | **57.30** | 84.82 | 82.91 | 80.24 | 78.31 |

## 5 CONCLUSION

In this paper, we certified the existence of the NSS phenomenon in SSM from the perspective of ETC theory and found that within multi states memories in state space equations significantly mitigated NSS. Building upon these theoretical analyses and findings, we proposed S4++, a simple, yet effective neural architecture for long sequence modeling. By leveraging the proposed SMR mechanism, S4++ effectively incorporates finer dependency biases, enabling efficient long sequence modeling. Furthermore, the complex inductive bias embedded in the cross-attention mechanism equips S4++ to effectively handle a wide array of data types. S4++ outperforms strong baselines across a range of sequence modeling tasks and data types, demonstrating significant improvements.

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
