# OpenReview forum: "S4++: Elevating Long Sequence Modeling with State Memory Reply"
_ICLR.cc/2024/Conference — ICLR 2024 Conference Withdrawn Submission_

### Official Review · Reviewer_rGQA · 2023-10-22

**Soundness:** 2 fair
**Presentation:** 1 poor
**Contribution:** 1 poor
**Rating:** 3
**Confidence:** 4

**Summary:**

This paper proposes the use of event-triggered control (ETC) theory to improve the stability of deep SSMs such as S4. The paper investigates how perturbations of the inputs can lead to non stable states (NSS) and develops conditions for when this happens. The paper then proposes the use of a convolution preprocessing step applied to the inputs which they refer to as a State Memory Replay (SMR) mechanism to address this issue. The combination of S4 and this SMR results in S4+. The paper further investigates the usefulness of combining attention with SSMs. Experiments are performed on toy tasks, language and LRA to illustrate the benefits of the proposed approaches.

**Strengths:**

- The idea to incorporate theory and ideas from control theory into the increasingly popular line of deep SSM work is an interesting and exciting addition to the literature in this area. In particular the promise of using these ideas to improve stability and memory are intriguing.

- The inclusion of the illustrative examples of the NSS issue are useful to help the reader better understand the issue in practice.

- The theoretical contributions appear to be correct (given the assumptions, though I have questions here, see below) though I only skimmed the proofs to check the arguments rather than carefully checking every detail line by line.

**Weaknesses:**

- The conclusions from Proposition 1 seemed to be based on the assumption that there is asymptotic instability of the states $x(t)$ due to the max eigenvalue $|\lambda_{max}|>1$. However, prior S4-related work (https://arxiv.org/pdf/2202.09729.pdf, https://arxiv.org/pdf/2206.11893.pdf)  suggest enforcing the left-half plane condition to ensure stability and most S4-related implementations have this as an option. I would expect this baseline to be cited, discussed and compared to in the main paper.
   - In the appendix of this submission, there is a remark suggesting that enforcing  $|\lambda_{max}|<1$ limits the long-term memory capacity of the model. But, I would then expect to see empirical evidence suggesting the proposed method has better long-term memory capacity than vanilla S4. As discussed below, the empirical results do not support this claim.
  - This remark would also seem to overstate the issue, since the role of the learnable timescale parameter in S4 is to extend out the exponential decay to help capture very long-range dependencies.

- The proposed SMR (multiplying a convolution of the input sequence with the input) seems to essentially be the short convolution/shift matrix and multiplicative gating proposed by H3 and Hyena (https://arxiv.org/abs/2302.10866). This connection should be discussed. While this makes the actual proposed method a less novel contribution, discussing this connection in more detail could add more theoretical insight into why this is a good thing to do in these other methods.

- The second contribution of the paper is the proposal to combine SSMs with Attention. However, this is not new and has been proposed before in MEGA (cited in the results but not discussed) and Block-state Transformer (https://arxiv.org/pdf/2306.09539.pdf). The architecture in Figure 5 looks very similar to the proposed architecture in Mega. The similarities and differences between S++ and this work should be discussed thoroughly and ablated.

- The motivation for the need of attention in this paper seems weak. Unless I am misunderstanding Section 3.2, the motivation seems to be the need for bidirectional modeling abilities while it is claimed S4 is simply a unidirectional model. However, almost all prior S4-related papers have proposed and used bidirectional SSMs. This is not to say it is necessarily incorrect that combining attention and SSMs is helpful, but the stated motivation and reasoning for why it is required seems incorrect.

- The empirical results are weak and do not help to support the claim that the proposed modifications lead to performance improvements. The LRA results exclude the most challenging Path-X task which many methods cannot improve over random guessing (this was one of the big contributions of S4). It is suspicious that this task was left out. Showing better performance on this task would also help to support the claim that the proposed method helps to improve the memory capacity over restricting the eigenvalues as mentioned above related to the relevant remark in the appendix.
   - Further, the reported LRA results seem to make no contact with prior reported results on this task. E.g. in this paper (https://arxiv.org/pdf/2206.11893.pdf) S4 has an average score much higher than the average score reported for all the methods in this paper. Similarly, in the Mega paper (a direct comparison to another SSM+attention based method).

- The presentation is poor with typos throughout. Further, while the Figures could be really helpful for the story, the resolutions of Figures 1 and 2 are too small, making it difficult to read and understand. Further, all Figures would benefit from having captions that explain the different components of the figures.

**Questions:**

1. Could you provide evidence that S4+ has better memory capacity than S4, as stated in the remark of the appendix after the proof of Proposition 1? One step would be to at least show that S4+ with eigenvalues that exceed magnitude 1 performs better than S4 with constrained eigenvalues less than 1 on the Path-X task (ideally using the recommended S4 hyperparameters from prior work). Perhaps some other synthetic memory tasks could be added as well. This point seems important to support the claims of the paper.

2. Along with Path-X, could you provide LRA results using the default hyperparameters for the baseline methods (e.g. using the S4 and Mega repos)? It is confusing to see very different reported results than what is reported elsewhere and makes the results hard to judge.

3. Am I misunderstanding the motivation of adding attention in Section 3.2? The motivation appears to be that because S4 is unidirectional it cannot solve cloze questions. Thus, it would then seem we need to add cross attention to solve these tasks. But prior S4-related papers show that bidirectional SSMs can model the dependencies mentioned in the first paragraph of Section 3.2. Again, I am not saying SSMs do not need attention, (e.g. see the motivation in the H3 paper), but the reasoning here seems incorrect.

4. There is at this point a growing line of SSM-related work. I would expect to see a related works section that discusses these other works and the connections with this paper. To name a few: GSS, H3, S4D, DSS, S5, Liquid-S4, MEGA, LRU, Hyena, Sgconv etc. I would expect these works (and others) to be cited and positioned with relation to this paper. In particular, GSS, H3 and Hyena are very related to the proposed SMR mechanism and MEGA and Block-state Transformers are very related to the proposed attention mechanism presented here.

5. I would encourage the authors to fix the resolutions of Figures 1 and 2 and to add explanatory captions to all figures.

Other suggestions or minor issues:
- Throughout the paper S4 is cited as Fu 2023 which refers to the H3 paper and not the correct citation for S4. Please correct this.
- Typo, in abstract, "even" should be"event"
- In section 2.1, the general SSM is introduced with general dimensions (similar to the presentation in S5 and LRU), however it is never mentioned that S4 uses a particular stack of 1-D filters. This could be confusing to readers when you jump to the convolution definition and dimensions below.

---

### Official Review · Reviewer_aFxd · 2023-10-30

**Soundness:** 1 poor
**Presentation:** 2 fair
**Contribution:** 1 poor
**Rating:** 3
**Confidence:** 4

**Summary:**

This paper tackles two problems in deep state space models.  Firstly, the authors derive an idea of non-stable states, where learned models are sensitive to imperceptible perturbations.  Secondly, convolutions and attention mechanisms are introduced to try and ameliorate these concerns.  Some theoretical and empirical results are presented.

**Strengths:**

I think combining different types of layers/techniques in deep sequence models is very valuable, and trying to build a “library” of intuitions about what pathologies are remedied by which layers/techniques is something that the community should begin to build.

**Weaknesses:**

Ultimately I do not believe this paper is close to publication-ready.  There may well be a reasonable contribution in this paper as it stands, but I have too many concerns about the pitching and evaluation of the idea to be confident in recommending acceptance. The paper itself is also littered with errors, which to me suggest that the paper was rushed and would greatly benefit from a round of revisions.

I will outline some more of my major concerns below (in no particular order).

## Major Concerns

*W.1.: Figures quoted*:  The figures quoted in your results table are incorrect (e.g. S4 should achieve an average of 86.09 as opposed to the 66.48 you quote).  The performance gains over bidirectional S4, Liquid S4, S5 etc are marginal at best against the incorrect numbers and are very below the “correct” numbers.  Please can the authors clarify?

*W.2.: Is NSS a problem?*:  If it is believed that there is jitter in the sampling timesteps of the system, then shouldn’t that be presented in the training data as well?  I am not convinced that the NSS problem is as significant as the authors make out.  Does adding some Gaussian noise (for continuous input spaces) to the input remedy the problem?  There is also no discussion of S5, which can handle irregularly sampled data (I understand you’re referring to “binned” data, but this is still a necessary point for discussion, even if it is qualitative).  Ultimately, I think it is not experimentally shown that NSS is actually a problem that appears outside of a single, _very_ tightly controlled experiment.

*W.3.: Unidirectionality of models*:  You say that S4 is unidirectional, but I don’t believe that is true.  S4/S4D/S5 all run an SSM in each direction, and hence gets bi-directional information.  You recover this through the attention mechanism in S4++, but I don't think comparing to unidirectional S4 models (either qualitative or quantitative) is fair.

*W.4.: Difference to MEGA*: My understanding is that MEGA (somewhat reductionist-ly) added an attention layer to an S4 model.  Can the authors clarify how their S4++ model is different from MEGA?  And why is there no discussion in the main text of MEGA?  Similarly, this introduction of attention re-introduces the quadratic complexity, meaning that the original motivation of S4 has been lost.  This is not mentioned in the paper.

*W.5.:  S4+ Motivation and grounding*: I just about follow through Theorem 1 that you are effectively looking to control the scale of the inputs to control the stability of the system.  (15) is similar to the result, sure, but how do you ensure that the constraints/requirements are met?  It would be nice to see somehow empirical validation that the extra complexity is acting as intended (could you compute the value of the message?), and that you’re not just adding extra computational flexibility in the system that is leading to marginally better performance.

## Minor weaknesses:
- You describe your method as both State Memory Replay and State Memory Reply.
- ETC theory is not discussed at all.
- Proposition 1 should include a proof sketch in the main paper.
- There is a litany of content/typographical errors, e.g.:
  - Only proper nouns should be capitalized.
  - Figures should be rendered as PDFs.
  - Is it EMC or ETC theory?
  - Fonts in figures should be about the same size as fonts in the body text.
  - Use the backtick (\`) when opening quotations (`Pythagorean theorem’ vs ‘Pythagorean theorem’).
  - Use \citet and \citep.
  - Citation style in the bibliography is inconsistent.

**Questions:**

I have no direct questions, but invite the authors to respond to the weaknesses outlined above.

---

### Official Review · Reviewer_1dtu · 2023-10-31

**Soundness:** 2 fair
**Presentation:** 1 poor
**Contribution:** 2 fair
**Rating:** 5
**Confidence:** 4

**Summary:**

- Introduces the ETC (even-triggered control) theory from controls that concerns the stability of state space models in relation to the input sampling. Through theory and synthetic experiments, the paper demonstrates a weakness with S4 models (which are designed for uniformly-sampled data) when they are deployed in variable-sampling regimes. This is termed the non-stable-states (NSS) phenomenon.
- Proposes a method SMR (state memory reply) for addressing NSS, which boils down to including a local convolution in the architecture, which is multiplied by the main inputs.
- Introduces S4++, a more sophisticated architecture incorporating SMR and cross-attention.
- Shows competitive results on autoregressive language modeling (WikiText-103) and long-range classification (Long Range Arena).

**Strengths:**

- The paper explores a setting and potential issue with prior state space models (SSMs) that has not been analyzed before: stability under sampling perturbations. Such a phenomenon is important to explore in data that is not sampled uniformly, as with many sequence modeling tasks.
- The synthetic tasks used to explore and illustrate this new phenomenon (Figure 1, 2, 4) are compelling and demonstrate the improvement of the proposed method.
- I find the introduction of the local convolutional modification (SMR) intriguing, as such techniques have been used empirically by many prior models (H3, Hyena, RWKV) but not theoretically explained.

**Weaknesses:**

1. The presentation has many small errors that make it more difficult to read.
2. The theoretical framework is not explained in a self-contained manner. For example, the ETC theory and Lyapunov functions involve two new parameters P and T, but it is not explained how these are related to the underlying SSM models which do not have these parameters.
3. Overall in the NSS part of the paper (the first main contribution), the connection between the theory (Theorem 1) and the proposed method (equation (15)) seems quite tenuous at worst, and insufficiently explained at best. While the proposed convolutional smoothing technique makes sense intuitively, and has been used by prior work empirically, it is not obviously related to the proposed theoretical motivation. I will raise my score if the connection from Theorem 1 to equation (15) is unpacked in more detail.
4. The second contribution of adding cross-attention is more marginal and has been done before in several related works.

**Questions:**

1. In the bidirectional GLUE experiments, how does S4++ compare with BiGS [1], another S4 extension for bidirectional sequence modeling?
2. How does the training setting for LRA differ from that of prior works? Many of the baselines (e.g. S4, Mega, H3) have reported numbers trained from the original released S4 repository. The numbers reported in Table 4 differ quite substantially from these standard evaluations.
3. I think it would be nice to explain the connections to related work more closely. For example, the first SMR idea seems very related to the local convolutions used in H3, Hyena, and RWKV. The cross-attention addition and overall architecture also seems closely related to Mega.

[1] Wang et al. "Pretraining Without Attention."

---

### Official Review · Reviewer_QwTp · 2023-11-01

**Soundness:** 2 fair
**Presentation:** 1 poor
**Contribution:** 3 good
**Rating:** 3
**Confidence:** 4

**Summary:**

The paper investigates state trajectory divergence due to fixed-grid discretization in SSM models (e.g., S4), and proposes an implicit (hypernetwork) parametrization for sampling points (SMR).

**Strengths:**

* The idea of parametrizing the step sizes and this implicitly controlling the discretization grid as a function of the data is interesting.
* I appreciated some of the examples that motivate shortcomings of naive discretization grids (e.g., Fig 2)

**Weaknesses:**

* The paper overlooks various SSM-based approaches that get around keeping $\Delta t$ fixed (for example, S5, which avoids the issue by computing the input-output via a parallel scan).
* The presentation can be significantly improved, some sections (3.2) could be greatly shortened.
*  The weakest part of the paper is by far section 3. I elaborate on my concerns in my questions below

**Questions:**

* Is your main point in 3.2 that SSMs are causal? There are many ways to address this in long convolution models (make the filters bidirectional, use an implicit representation a la CKConv, FlexConv or Hyena).
* Have you explored other parametrizations for the sampling points, e.g. learning $\Delta t$ directly? I understand a convolution makes sense because it can be evaluated efficiently, but I am curious about other variants. This would be a strong ablation to add to the paper.
* What is the motivation behind rescaling only the inputs $u$ with SMR, other than preserving the convolution form of the system? Why not learn the step size of the discretized system also?
* Language (or LRA, for that matter) is not the best place to showcase adaptive $\Delta t$. The paper could benefit from showing large gains from a domain that really benefits from models capable of adapting to irregular samples.
* S4++ in 3.3 is a mixture of various existing blocks (H3, Hyena, BiGS); the main idea is to mix a SSM, gating and attention. My concern is that this design is somewhat arbitrary; it is not motivated by anything other than experimental results, which are strange. For example, 25 perplexity on Wikitext for models of that size is really bad; GLUE results are similarly poor, with tasks (like RTE) that are off by more than 5 points from a good BERT baseline. This in my mind invalidates any claim on the performance of the model.

This is unfortunately a paper with an interesting idea that is brought down by poor presentation and evaluation. Most interesting design choices surrounding what I see as the key contribution (SMR) are not sufficiently explored.